Analysis of microbiota in elderly patients with Acute Cerebral Infarction

Huang Lin
Wang Teng
Wu Qian
Dong Xin
Shen Feifei
Liu Dong
Qin Xiaoxuan
Yan Lanyun yanlanyun@126.com
Wan Qi wanqi@jsph.org.cn
Department of Neurology, The First Affiliated Hospital of Nanjing Medical University , Nanjing , Jiangsu Province , China
Fiorentini Carla
Electronic publication date: 2019 Jun 12
Publication date: 2019
Volume: 7
Electronic Location ID: e6928
Received 2018 Oct 10; Accepted 2019 Apr 3
Copyright: ©2019 Huang et al.
Copyright year: 2019
Copyright holder: Huang et al.
License: This is an open access article distributed under the terms of the Creative Commons Attribution License, which permits unrestricted use, distribution, reproduction and adaptation in any medium and for any purpose provided that it is properly attributed. For attribution, the original author(s), title, publication source (PeerJ) and either DOI or URL of the article must be cited.
License URL: https://creativecommons.org/licenses/by/4.0/

Keywords: Acute Cerebral Infarction, Blautia obeum, Prevotella copri, Streptococcus infantis, Microbiota

Funding: National Natural Science Foundation of China 81600970 This work was supported through funding from the National Natural Science Foundation of China (NSFC 81600970). The funders had no role in study design, data collection and analysis, decision to publish, or preparation of the manuscript.

==============================
Background and Aims

Recent evidence suggest that microbiota is associated with almost all major types of diseases, including cardiovascular diseases. However, its role in Acute Cerebral Infarction remains unexplored. It is important to understand the diversity and distribution of gut microbiota (GM) in patients with Acute Cerebral Infarction and the role that GM plays in this type of disease.

Methods

We performed pyrosequencing on the gut microbiota of 40 individuals in order to elucidate whether the composition of the microbiota differs between patients with Acute Cerebral Infarction and healthy controls: Of these individuals, there were 31 with Acute Cerebral Infarction and nine controls. We applied linear regression to calculate the correlation between the gut flora and disease risk factors. Finally, KEGG functional enrichment analysis was conducted to examine the correlation between the gut flora and Acute Cerebral Infarction.

Results

The overall microbial structure was similar in both the controls and the patients, but the control group had higher relative presence of Blautia obeum while the presence of Streptococcus infantis and Prevotella copri were relatively higher in the patient group. Using linear regression, we found that Blautia obeum was negatively associated with white blood cell count and Streptococcus infantis was positively correlated with creatinine and lipoprotein. The KEGG pathway analysis indicated that the bio-pathways including methane metabolism, lipopolysaccharide synthesis, bacterial secretion, and flagellar assembly of the gut microbiota in the patient group was expressed differently than that of the controls. We identified three differentially expressed gut microbial functions in Acute Cerebral Infarction and found four bacterial pathways that might be related to the development of this disease.

Conclusions

Our study identified three abnormally-expressed bacteria—Blautia obeum, Streptococcus infantis, and Prevotella copri—in patients with Acute Cerebral Infarction compared with healthy controls. It reveals a correlation of these bacterial species with Acute Cerebral Infarction as they relate to disease factors and functional pathways. These findings may shed light on the treatment of cerebral infarction because gut microbiota could serve as a potential therapeutic approach for the treatment of cardiovascular and metabolic diseases.

Introduction

Stroke causes 6.2 million deaths globally (Esenwa & Gutierrez, 2015), making it the second leading cause of death. Cerebral infarction, also known as cerebral stroke, accounts for 87 percent of all stroke cases (Roger et al., 2011), and occurs when the arteries that supply blood to the brain are blocked or narrowed so that blood flow is interrupted. The main cause of cerebral infarction is atherosclerosis, which results from the development of fatty deposits in the lining of the vessel walls. Tissue plasminogen activator (TPA) (Raphaeli et al., 2015) is a common treatment for cerebral infarction, but many patients do not arrive at the hospital within the limited time needed to effectively access brain-saving treatments. Thus it is critical to identify a stroke and to seek treatment immediately to have the best possible chance of a full recovery.

Dysbiosis of the gut microbiota has not only been shown to contribute to the development of the immune system (Butto & Haller, 2016), but also, remarkably, to the development of the Central Nervous System (CNS) (Borre et al., 2014). Gut microbiota is related to the main systems of the human body, such as the immune system, nervous system, endocrine system, digestive system, respiratory system and even the circulatory system. Atherosclerosis and stroke are two major diseases of the circulatory system. The major culprit in these diseases is trimethylamine oxide (TMAO) (Wu et al., 2018). The formation of TMAO is closely related to the metabolism of intestinal flora. This information leads to the question of whether the gut microbiota in patients with Acute Cerebral Infarction is different from those without cerebral infarction. Furthermore, it is possible that different microbiota might be further related to other indicators of disease. The above hypothesis can further elaborate the theory of a brain-gut axis and may provide a theoretical basis for the prevention and treatment of Acute Cerebral Infarction. Researchers have identified variation in the composition of gut microbiota in stroke patients (Houlden et al., 2016; Singh et al., 2016; Yin et al., 2015) but only a few experimental studies have been published which focus on the role of gut microbiota in cerebral infarction. The link between gut flora and Acute Cerebral Infarction remains to be established, so the exact mechanisms of how gut microbiota are involved in cerebral infarction are yet to be determined.

To elucidate this relationship, we conducted the described study in which 40 samples (31 from patients with Acute Cerebral Infarction and nine from controls) were analyzed for 16S rRNA genes by pyrosequencing to detect differences in the microbial composition of the gut and to explore the correlation between gut flora and Acute Cerebral Infarction using linear regression and KEGG enrichment analysis.

Methods and Results

Methods

Sample collection

Fecal samples were collected from patients, whose median age was 61 years old, with Acute Cerebral Infarction from the First Affiliated Hospital of Nanjing Medical University, from February 2018 to May 2018. There was no negative control for age or gender groups; instead, the range was set from 40 to 80 years. The main inclusion criteria for the study group was Acute Cerebral Infarction as diagnosed by a neurologist. All patients underwent Magnetic Resonance Imaging (MRI) or a Computed Tomography (CT) scan to assess ischemic lesions. The large-artery atherosclerotic subtype of atherosclerosis was determined according to the TOAST classification system. Radiological evidence of intracerebral hemorrhage, a definitive cause of stroke or TIA, is not associated with atherosclerosis (e.g., carotid dissection, perivascular programmed stroke, cardiac stroke, or other TOAST subtypes), serious comorbidity, or a medical condition within one month prior to admission (i.e., patients with congestive heart failure, respiratory failure, renal failure, or severe liver dysfunction), or those taking probiotics or antibiotics, were excluded from the study group. The control group consisted of asymptomatic participants (according to a physical examination and self-report of no acute disease). The control group excluded participants who had taken antibiotics or probiotics within a month prior to admission, or who had a history of cerebrovascular disease. In addition, we excluded interference from diabetes, coronary heart disease, and renal dysfunction. Color Doppler imaging and transcranial Doppler ultrasound were performed in all control groups to determine cardiovascular status. The trial and informed consent was approved by the ethics committee of the First Affiliated Hospital of Nanjing Medical University (2018-SRFA-122).

Fresh stool samples were collected in the morning to avoid surface contamination and urine contamination. Internal stool samples of 3–5 g were collected with a clean sterile spoon and were placed in a sterile, airtight tube (QIAGEN). The sample was preserved immediately in an anaerobic environment in a liquid nitrogen tube in order to avoid repeated freezing and thawing. After transportation on dry ice, the sample was stored at −80 °C. The fecal stool collection in the patient group was completed within 48 h of admission.

Detection of biochemical specimens

All fasting venous blood samples of about 5 mL were collected into the drying tube and then shot through the barrel to the pretreatment room. After the barcode was brushed in the pretreatment room, the samples were put into the Beckman Coulter assembly line. On the assembly line, each specimen was centrifuged for 7 min at 3,000 rotations. The centrifuged specimen was detected by the Beckman Coulter AU5800 biochemical analyzer (Beckman Coulter, Inc, Brea, CA, USA). Different testing reagents were used according to the different samples being tested (the reagents needed for related testing items are in Table S1).

Routine blood test

The 2 mL venous blood samples, which were fasting samples taken early in the morning, were collected into the EDTA anticoagulant tube without centrifugation and entered the Beckman Coulter pipeline (Unicel DxH 800).

PCR amplification and pyrosequencing of bacterial 16S rRNA genes

Bacterial DNA was extracted from fecal samples according to the manufacturer’s instructions using the Fecal DNA Extraction Kit (QIAGEN). The V4 variable region of the bacterial 16S rRNA gene was amplified by polymerase chain reaction (PCR) using the bar code primers 514F (GTGCCAGCMGCCGCGTAA) and 805R (GGACTACHVGGGTWTCTAAT). The PCR cycle conditions were: initial denaturation at 94 °C for two minutes; denaturation at 94 °C for 30 s; denaturation at 52 °C for 30 s; denaturation at 72 °C for 45 s; and extension at 72 °C for five minutes. Each 25 µL reaction consisted of 0.5 µL of template DNA, 2.0 µL of dNTP mix (2.5 mmol/L; Takara), 2.5 µL of Takara 10 Ex Taq buffer (Mg2+ free), 1.5 µL of Mg2+ (25 mmol/L), 0.25 µL of Takara Ex Taq DNA polymerase (2.5 units), 0.5 µL of 10 µmol/L bar code primer 514F, 0.5 µL of 10 µmol/L primer 805R, and 17.25 µL of double-distilled water. All dilution was carried out with sterile double distilled water.

All PCR products were combined and sent to Beijing Genomic Institute for sequencing using the Illumina Miseq (PE 150), according to the manufacturer’s protocol (Zhou et al., 2011).

Statistical analysis

The tag number for each taxonomic rank (species) or OUT (operational taxonomic units) in the samples was summarized in a profiling table or histogram, drawn with R (v3.1.1) software. The nonparametric Wilcoxon Rank-Sum test was used to identify OTU among different groups in QIIME. In order to determine the differences between the two groups not reported in QIIME, the OTUs of P < 0.1 in QIIME analysis were classified into generic levels and then statistical tests were implemented using Graphpad prim6. A chi-square test was used for categorical variables using SPSS. A value of P < 0.05 was considered statistically significant in the comparison groups.

Results

Composition of gut flora in patients with Acute Cerebral Infarction and in controls

In this study, a total of 40 blood and fecal samples were collected from 31 patients and nine healthy controls. Clinical characteristics of the whole population are shown in Table 1. The rates of previous HBP (p = 0.005) and previous HLP (p = 0.036), as well as the levels of TC (p = 0.028), HDL (p = 0.002), TG (p = 0.003), WBC (p = 0.023), and LP (a) (p = 0.003) were significantly higher in the patients with Acute Cerebral Infarction than in the control subjects. In addition, there was no difference in GLU (p = 0.178), Cr (p = 0.13) and UA (p = 0.134) between the two groups, which indicated that the patients sampled were significantly free of diabetes and renal dysfunction Table 2.

Table 1 Characteristics of the study participants.

Variable	Patients	Controls	P Value	
N	31	9	–	
Male, n (%)	22 (70.97)	6 (66.67)	0.385a	
Age (Median, IQR)	61 (40-94)	61 (53-69)	.238b	
Current smoker(%)	19 (61.29)	0	0.001a	
Previous diabetes, n (%)	0	0	–	
Previous CAD, n (%)	0	0	–	
Previous HBP, n (%)	23 (74.19)	2 (22.22)	0.005a	
Previous HLP, n (%)	11 (35.48)	0	0.036a	
Previous ischemic stroke(%)	1 (3.23)	0	0.585a	
Notes.

CAD coronary artery disease

HBP high blood pressure

HLP hyperlipidemia

LAA large-artery atherosclerosis

Continuous variables are presented as medians (interquartile range). Significant differences between groups were analyzed with the:

a Chi-square test

b Kruskal Wallis Test

Table 2 Blood Biochemistry Results.

Variable	Patients	Controls	P Value	
TC, mmol/L	4.984 ± 1.669	4.089 ± 0.624	0.028a	
HDL, mmol/L	1.056 ± 0.268	1.437 ± 0.268	0.002a	
LDL, mmol/L	3.199 ± 1.068	2.712 ± 0.548	0.199a	
TG, mmol/L	1.93 ± 1.239	0.961 ± 0.268	0.003b	
GLU, mmol/L	5.132 ± 0.645	4.646 ± 0.590	0.178a	
WBC, 109/L	6.733 ± 1.733	4.932 ± 1.456	0.023a	
Cr, µmol/L	75.28 ± 18.70	59.52 ± 8.443	0.13a	
UA, µmol/L	329.4 ± 93.95	277.9 ± 66.79	0.134b	
LP(a), mg/L	325.8 ± 262.9	74.89 ± 38.72	0.003c	
Notes.

LDL low-density lipoprotein

TC total cholesterol

TG triglycerides

GLU glucose

WBC white blood cell count

Cr creatinine

UA uric acid

LP(a) Lipoprotein(a)

Continuous variables are presented as means ± standard deviations or as medians (interquartile range). Significant differences between groups were analyzed with the:

a Kruskal Wallis Test

b Student’s t test

To investigate whether gut microbiota composition differed between the study population of patients with cerebral infarction and healthy controls, we performed sequencing of the V4 region of the 16S ribosomal RNA gene from fecal samples. After filtering for quality, a total of 1,670,914 reads were included for downstream analysis and an average of 41,772 ± 293 sequences (SD) were assigned to each sample.

PLS-DA (Partial least squares discrimination analysis) was performed in order to further distinguish between the groups. This was achieved by rotating PCA (Principal Component Analysis) components such that a maximum separation between classes was obtained and to understand which variables carried the class separating information. The PCA components were calculated to analyze the composition of the gut microbiota in patients with Acute Cerebral Infarction and in the control group (Fig. 1A). However, the diversity of species richness (represented by Chao, observed species, ace), richness and evenness (represented by Shannon and Simpson indexes) of the microbial community indicated, and the number of community species (Good’s coverage) revealed that no significant difference in gut flora diversity could be identified between these two groups (Figs. 1B–1G).

Figure 1 Comparison of a-diversity between the gut microbiota of patients and controls.

(A) OTU Based PLS-DA Analysis. Orange triangles represent samples (intestinal microbiota) from patients; blue circles represent samples of controls. Five indices were used to represent the a-diversity between the patients and controls (B, observed species; C, chao; D, ace; E, shannon’s diversity; F, simpson diversity); G, The number of community species between the patients and controls. N indicates healthy people and P indicates patients.

Figure 2 Analysis of the variation of intestinal gut in species level incontrols and Acute Cerebral Infarction patients.

(A) The relative abundance of each species in the intestinal gut in each sample from Acute Cerebral Infarction patients and controls (B indicates healthy people and A indicates patients); The average relative abundance of three gut microbiota (B, Blautia boeum; C, Streptococcus infantis; D, Prevotella copri) between patient group and control group. P < 0.05 was considered statistically significant in the comparison groups, N indicates normal people and P indicates patients. All the above analyses are T-test.

Gut bacterial species variations in Acute Cerebral Infarction patients and controls

In contrast, the different taxa abundance (Class, Family, Genus, Order, Phylum and Species) (Figs. S1–S5 and Fig. 2A) in the gut microbiota showed significant differences at the species level between patients with Acute Cerebral Infarction and the healthy controls. The bacterial species with the highest relative abundances were Blautia obeum (p = 0.0441), Streptococcus infantis (p = 0.017), and Prevotella copri (p = 0.0099). The most remarkable difference was found in Blautia obeum. The presence of Blautia obeum is relatively lower in the patient group while Streptococcus infantis and Prevotella copri are relatively higher in cerebral infarction patients compared to the control group (Figs. 2B–2D).

Correlation between gut microbial taxa at the species level and Acute Cerebral Infarction

After identifying three abnormally-expressed gut microbiota species, we analyzed whether there were any associations between bacteria and cardiovascular risk factors using linear regression. Findings are indicated in Table 2. They show that Blautia was negatively associated with the white blood cell count (r2 = 0.1053). In contrast, Streptococcus showed a positive correlation with creatinine (r2 = 0.1328) and lipoprotein (r2 = 0.1004), whereas Prevotella had no significant relation to any disease risk factors.

To predict the abundance of gene families and the related functional pathways of microbial communities in the fecal contents, KEGG functional pathway analysis, a predictive metabolism approach, was performed based on the 16S rRNA gene sequencing and Green Genes database (Fig. 3). Results suggested that many bacterial pathways involved in methane (p = 0.0103) (Fig. 3B), lipopolysaccharide (p = 0.0166 and p = 0.0249) (Figs. 3C and 3F), secretion (p = 0.0156) (Fig. 3D) and flagella (p = 0.0065) (Fig. 3E) functions were significantly modulated in Acute Cerebral Infarction.

Figure 3 Pathway abundance analysis of microbial taxa with KEGG.

(A) KEGG functional pathway abundance analysis of the gut microbiota (B indicates healthy people and A indicates patients); Average pathway abundance of a gut bacterium (B, Methane_metabolism; C, Lipopolysaccharide_ biosynthesis_ proteins; D, Secretion_system; E, Flagellar_assembly; F, Lipopolysaccharide_biosynthesis) between patient group and control group. All the above analyses are T-test.

Functional analysis showed that the methane metabolism of these three bacteria was present in patients with Acute Cerebral Infarction. But lipopolysaccharide synthesis was enhanced in patients with Acute Cerebral Infarction. In addition, acute cerebral infarction may be positively correlated with bacterial secretion and flagellar assembly in these three gut microbiota.

Discussion

Our results support the concept of bidirectional communication along the brain-gut axis. Recent reports have shown that microbial populations play a significant role in development of cerebral infarction (Cryan & Dinan, 2012; Yin et al., 2015), and that diseases caused by antibiotic treatment can affect the outcome of stroke (Benakis et al., 2016). At present, bacteria have been considered as a pathogenic factor in the development of cardiovascular diseases. The difference of intestinal flora between atherosclerotic patients and unaffected people suggests that the change of intestinal flora may be related to atherosclerosis and that intestinal flora may change the metabolites of some substances, such as, TMAO (Yin et al., 2015). Studies have shown that increased levels of TMAO are associated with an increased risk for major cardiovascular and cerebrovascular incidents. TMAO can be used as an accurate screening tool for predicting the future risk for heart attack, stroke, and death among people not recognized by traditional risk factors and blood tests (Tables 1 and 2). Here, we report that Acute Cerebral Infarction itself significantly affects the intestinal microbial composition and that these changes may be associated with the occurrence and development of Acute Cerebral Infarction.

The intestinal microbiome has been further implicated in the pathogenesis of multiple diseases such as obesity, depressive disorders, chronic ileal inflammation, liver disease, and atherosclerosis (Chang et al., 2015; Fei & Zhao, 2013; Koeth et al., 2013; Le Roy et al., 2013; Llopis et al., 2016; Schaubeck et al., 2016; Zheng et al., 2016). For example, the metabolism of dietary L-carnitine, a nutrient in red meat, by intestinal microbiota was demonstrated to promote atherosclerosis and lead to cardiovascular disease risk by producing trimethylamine and trimethylamine-N-oxide (Koeth et al., 2013). However, the role of the intestinal microbiota in cerebral infarction, the development of diabetes, renal dysfunction, and other related diseases is not well understood (Ji et al., 2017).

To address this void, the present study sought to determine whether the composition of the gut microbial community differed between Acute Cerebral Infarction patients and healthy controls, and whether the microbial taxa of the gut correlates with risk factors for cerebral infarction. In terms of microbiota composition, no significant difference was identified between the patient group and the control group (Fig. 1). However, the control group showed a higher prevalence of Blautia obeum (0.315%) in the gastrointestinal system than the disease group (0.115%) (Fig. 2).

Blautia obeum is a species of anaerobic, gram-positive bacteria found in the gut and is recognized as being dominant in the human colon (Lawson & Finegold, 2015). It has been reported that B. obeum, along with other relevant taxa, play an important role both in the recovery process from V. cholerae infection and the maturation of microbiota in children (Hsiao et al., 2014). There has been speculation about the possibility that some of these bacteria may be helpful in the repair of gut microbiota in individuals whose gut communities have been wounded through a variety of insults, including enteropathogen infections (Hsiao et al., 2014). In addition, Hatziioanou et al. (2017) identified and characterized a gene cluster from the human gut isolate Blautia obeum A2-162 which encodes the novel nisin-like peptides NsoA1-3 and NsoA4. Moreover, the antimicrobial activity of the host strain could be detected in the presence of trypsin. Thus, based on our discovery, B. obeum may function as an anti-stroke factor and therefore its role in Acute Cerebral Infarction deserves a thorough investigation.

We also found that both Streptococcus infantis and Prevotella copri were expressed in greater quantity in Acute Cerebral Infarction patients than in healthy people. Streptococcus infantis is reported to cause minocycline resistance; its genetic basis is due to tet (S) present on a novel low copy number plasmid flanked by IS1216 elements, which likely mediate its excision (Ciric et al., 2014). Prevotella are members of the oral, vaginal, and gut microbiota and are often recovered from anaerobic infections of the respiratory tract. Overgrowth of Prevotella in other diseases is known to occur in cases, such as in hypertension (Li et al., 2017) and chronic inflammatory disease (Larsen, 2017). When compared with strict commensal bacteria, Prevotella exhibits increased inflammatory properties and studies indicate that Prevotella predominantly activates the toll-like receptor 2, leading to the production of Th17-polarizing cytokines by antigen presenting cells, including interleukin-23 (IL-23) and IL-1. Furthermore, the expansion of Prevotella copri is associated with enhanced susceptibility to arthritis, and the colonization of mice with the bacteria revealed the ability of P. copri to dominate the intestinal microbiota, resulting in an increased sensitivity to chemically induced colitis (Scher et al., 2013). There is growing evidence linking Prevotella copri to human diseases (Scher et al., 2013). Our study revealed that these two bacteria were found in relatively higher abundance in those with Acute Cerebral Infarction than in unaffected patients; whether they exert an indirect influence on the pathogenesis of this disease remains to be determined.

The correlation between gut flora and disease risk factors was also analyzed using linear regression. The data revealed a negative correlation between Blautia’s and WBC (r2 = 0.1053, P = 0.04), and a positive correlation between Streptococcus, creatinine (r2 = 0.1328, P = 0.022) and lipoprotein (r2 = 0.1004, P = 0.0494) (Table 3). Combined with previous findings, it may be concluded that Blautia may support the immune system, while Streptococcus may contribute to the development of Acute Cerebral Infarction.

Table 3 Correlation between gut microbial taxa at the Species level and disease risk factors.

	TC	HDL	LDL	TG	GLU	WBC	Cr	UA	LP(a)	
Blautia	N.S	N.S	N.S	N.S	N.S	P = 0.04, r2 = 0.1053−	N.S	N.S	N.S	
Prevotella	N.S	N.S	N.S	N.S	N.S	N.S	N.S	N.S	N.S	
Streptococcus	N.S	N.S	N.S	N.S	N.S	N.S	p = 0.0226, r2 = 0.1328+	N.S	p = 0.0494, r2 = 0.1004+	
Notes.

P-values and r2-values for linear regression. A + or − indicates positive or negative association, respectively. N.S indicates not significant.

There are reports (Brzosko, Szkolka & Mysliwiec, 2009; Mekhlafi, Ibrahim & Rayyis, 2018) that impaired renal function is a strong risk factor for cardiovascular disease with a poor prognosis for the affected patients. Renal insufficiency can predict the long-term mortality after Acute Cerebral Infarction. However, the in-hospital mortality after Acute Cerebral Infarction is also closely related to the disturbance of consciousness, the severity of the cerebral infarction, body temperature, blood sugar, and other complications. Cr and UA are the main indicators of renal function. Although there is not much difference between the normal group and the disease group, there is a positive correlation between serum Cr and Streptococcus in the disease group, so it is speculated that Streptococcus may affect the kidney, which further influences the development of cardiovascular disease.

According to the report, we found that the key residues of lipoprotein PiaA, trp158, stabilized the iron–chromium complex in Streptococcus pneumonia (Zhang et al., 2017). There is a need for further study of the idea that Streptococcus may form a complex with the lipoprotein and consequently promote the increase in the lipoprotein.

Finally, we compared the differences of microbial bio-pathway enrichment between the patient group and control group using KEGG enrichment analysis. The analysis revealed that methane metabolism was inhibited in patients with Acute Cerebral Infarction, while lipopolysaccharide synthesis (p = 0.0166 and p = 0.0249), secretion pathways (p = 0.0156) and the flagellar assembly process (p = 0.0065) were enhanced in these patients (Fig. 3).

In addition, due to the limitation of the sample size, the correlation of the results may not be particularly strong, so increasing the sample size to determine the accuracy of the results of this experiment and later clinical guidance ought to be completed. The metabolites of bacteria have not been tested, but a series of deviations caused by bacterial changes can only be predicted by bioinformatics. In subsequent experiments, we need to further explore the changes of microbial metabolites such as TMAO, short-chain fatty acids, and so on, in order to confirm the pathway of microbial changes affecting cerebral infarction.

In conclusion, this study identified three abnormally-expressed bacteria, Blautia obeum (p = 0.0441), Streptococcus (p = 0.017) and Prevotella (p = 0.0099) in patients with Acute Cerebral Infarction compared with healthy controls. It also revealed a correlation of these bacterial species to Acute Cerebral Infarction as they relate to disease factors and functional pathways. These findings may shed light on the treatment of Acute Cerebral Infarction because gut microbiota could serve as a potential therapeutic approach for the treatment of cerebrovascular and metabolic diseases.

Supplemental Information

Table S1 Reagent names and manufacturers

Click here for additional data file.

Figure S1 Analysis of the variation of intestinal gut in Class level in controls and Acute Cerebral Infarction patients

The relative abundance of each Class in the intestinal gut in each sample from Acute Cerebral Infarction patients and controls (A indicates patients and B indicates healthy people).

Click here for additional data file.

Figure S2 Analysis of the variation of intestinal gut in Family level in controls and Acute Cerebral Infarction patients

The relative abundance of each Family in the intestinal gut in each sample from Acute CerebralInfarction patients and controls (A indicates patients and B indicates healthy people).

Click here for additional data file.

Figure S3 Analysis of the variation of intestinal gut in Genus level in controls and Acute Cerebral Infarction patients

The relative abundance of each Genus in the intestinal gut in each sample from Acute CerebralInfarction patients and controls (A indicates patients and B indicates healthy people).

Click here for additional data file.

Figure S4 Analysis of the variation of intestinal gut in Order level in controls and Acute Cerebral Infarction patients

The relative abundance of each Order in the intestinal gut in each sample from Acute CerebralInfarction patients and controls (A indicates patients and B indicates healthy people).

Click here for additional data file.

Figure S5 Analysis of the variation of intestinal gut in Phylum level in controls and Acute Cerebral Infarction patients

The relative abundance of each Phylum in the intestinal gut in each sample from Acute CerebralInfarction patients and controls (A indicates patients and B indicates healthy people).

Click here for additional data file.

Additional Information and Declarations

Competing Interests

Author Contributions

Human Ethics

DNA Deposition

Data Availability

The authors declare there are no competing interests.

Lin Huang performed the experiments, analyzed the data, prepared figures and/or tables.

Teng Wang performed the experiments.

Qian Wu, Xin Dong and Feifei Shen analyzed the data.

Dong Liu and Xiaoxuan Qin contributed reagents/materials/analysis tools.

Lanyun Yan conceived and designed the experiments, contributed reagents/materials/analysis tools.

Qi Wan conceived and designed the experiments, authored or reviewed drafts of the paper, approved the final draft.

The following information was supplied relating to ethical approvals (i.e., approving body and any reference numbers):

The trial was approved by the ethics committee of the First Affiliated Hospital of Nanjing Medical University (2018-SRFA-122).

The following information was supplied regarding the deposition of DNA sequences:

The 16S rRNA sequences described here are accessible via Sequence Read Archive accession numbers SRR8525598 to SRR8525637.

The following information was supplied regarding data availability:

Data is available at Wan (2019): results.rar. figshare. Dataset. DOI: 10.6084/m9.figshare.8081432.v1.

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
