# Peer review of "Analysis of microbiota in elderly patients with Acute Cerebral Infarction"

_PeerJ, doi:10.7717/peerj.6928_

## Round 0.1 · original submission · Major Revisions

The present investigation, although potentially interesting, presents several criticisms that require a deep revision, before being considered for acceptance. All reviewers' concerns should be addressed.

Reviewer 1 ·

Basic reporting

This study by Huang et al aims to identify microbiota changes in patients with acute cerebral infarction so as to identify a causal relationship. Despite a relevant attempt, there are several discrepancies associated with the manuscript:

Authors argue that there are no published reports identifying a causal link between Cerebral Infarction and microbiota (Line 171). As far as this reviewer can tell, a recent report has identified several microbial changes in the same line of work (Ref: MOLECULAR MEDICINE REPORTS 16: 5413-5417, 2017) which is in sharp contrast to work presented in this paper. In this reviewer’s opinion, authors need to discuss the significance of their results in wake of any relevant published work, especially as stated above.

Lines 188 and 200 directly contradict each other. Please proof read.

Experimental design

There is no mention of how the blood biochemistry assays were performed. Authors have neither described the results of variations in the blood tests (provide a separate bar diagram and not in Table 1) and nor discussed any statistical changes within the normal and disease groups. As per Table 1, it seems a significance test is applied but it is not clear. For instance, in Table 1, the parameter Cr has p value of 0.13 which means that the difference is non-significant. This implies that the parameter is not a marker of the disease and thus finding a correlation would be meaningless. Please address this issue.

The statistics applied is not clear. For instance, Table 1 suggests three different significance tests were performed. Why so? Also, while reporting values please mention the relevant SD or SEM and not the range. Preferably, report with a bar diagram or a box-whisker plot.

I also have a concern with the huge variation in sampling numbers. The control group has only 9 subjects while as many as 31 subjects of cerebral infarction were considered. This could skew the data interpretation. For instance, abundance of Blautia in both normal and patients seems to have a similar range, despite the observed significant change. Please elaborate this in relation to the chosen statistical method.

The data on correlation between bacteria taxa and presumed risk factors is very crude. The correlation observed is very poor as R2 value is closer to 0/0.1 and hence doesn’t appear to be very significant. In general, the blood tests and correlation performed render very little conclusive data. Please discuss.

State the median age of sample subjects in the methods section.

Split Table 1. Only include subject characteristics, while blood biochemistry results should be reported separately.

Validity of the findings

Please discuss a causative link between gut dysbiosis and how it could actually promote risk of infarction to emphasize your results.

While discussing results, please state any changes observed in terms of percent increase or decrease preceding with statistical significance and p value.

Please state a hypothesis of your work in the introduction section and how you plan to check it. Then based on your observations, discuss whether you accept or reject your hypothesis in the discussion section.

State limitations of the study in a separate paragraph.

Please state Ethics committee approval number with date.

Reviewer 2 ·

Basic reporting

no comment

Experimental design

no comment

Validity of the findings

no comment

Additional comments

The paper on analysis of microbiota in patients with Acute Cerebral Infarction is interesting. The authors have analyzed the contribution of gut microbiota (GM) in the development of acute cerebral infarction (ACI). Authors compared the GM and blood samples of 9 healthy individuals and 31 individuals with ACI and demonstrated that (1) abundances of Blautia obeum, Streptococcus infantis and Prevotella copri play role in the development of ACI, (2) a negative correlation between Blautia and WBC, and a positive correlation between Streptococcus and creatinine and lipoprotein and, (3) desired biological pathways in GM of patients with ACI are enriched.
Though, this study claims that B. obeum, S. infantis and P. copri play role in development of ACI, I find results unconvincing. Sample size is small. In figure 2B, it seems that if number of healthy persons will be increased, there will not be any significant difference in the abundance of B. obeum between healthy individuals (N) and patients (P). Similarly, for the abundance of S. infantis and P. copri, several patients have relative abundances similar to the healthy individuals. Also figure 3B does not show obvious differences in the average pathway abundance between N and P. These scenarios make me doubtful to say that these bacteria can influence the development of AIS. Along this line, authors also claim that overall composition of GM is similar in healthy individuals and patients, and P. copri does not have any relation to any disease risk factors. What we see here in results could result from the differences in dietary habits and age. It is advised that manuscript should report and interpret above facts, and authors should avoid from making any solid claims.

Minor comments:
1. Introduction needs to be more descriptive.
2. Line 126: define OTU.
3. Line 130: define PLS-DA.
4. Line 145: correct (Figure 2.)
5. Line 155: correct (Figure. 3)
6. Line 158: change “inhabited” to “present” or “prevalent” or as needed.
7. Line 179: correct reference style “colon[16].”
8. Line 200: it seems that “relatively low abundance” should be “relatively high abundance”.
9. Mention following in full when they appear first time in the manuscript: Blautia obeum, Streptococcus infantis, Prevotella copri, acute ischemic stroke.
10. Change Acute Ischemic Stroke to Acute Cerebral Infarction in the text, which would be consistent with the manuscript title.

---

## Round 0.2 · Minor Revisions

In the revised manuscript, the Authors have addressed all the concerns raised by the reviewers. However, before acceptance, the English should be carefully revised and also the few minor points indicated by the reviewers (see below) should be taken into account.

Reviewer 1 ·

Basic reporting

Authors have responded to earlier issues. However, there are several grammatical errors that need to be corrected before publication.

Experimental design

The median age of subjects is reported as 61 years. I advise authors to incorporate the word 'elderly/aged' in the manuscript title so as to accurately reflect the study context.

Validity of the findings

No comment

Additional comments

Authors have improved the manuscript based on suggested changes. Further changes recommended are general in nature.

Reviewer 2 ·

Basic reporting

no comment

Experimental design

no comment

Validity of the findings

no comment

Additional comments

Authors have justified my previously raised issues. The revised manuscript has numerous typos and I suggest that authors should thoroughly re-review the manuscript for typos.

Minor corrections
1. At multiple places genus and species are not italic.
2. Genus and species should be written in full when they appear first time, and afterword genus should be abbreviated.
3. Change “Lipoprotein” to “lipoprotein”, when it appears in the middle of a sentence.
4. Line 102: Full stop is not properly spaced at the end of sentence “…coronary heart disease and renal dysfunction”.
5. Lines 107-108: air tight should be airtight.
6. Line 120. Properly place the comma at “The venous blood samples”.
7. Lines 130-131: TaKaRa should be Takara.
8. Line 142: Full stop is not properly spaced at the end of sentence “…categorical variables using SPSS”.
9. Line 152: “dysfunctionin” should be “disfunction”.
10. Line 239: “leading to production of” should be “leading to the production of”.
11: Line 243: Space after full stop “2013).There is growing evidence”.
12: Line 199: trimethylamine oxide should be TMAO.
13: Line 273: MTAO should be TMAO.

---

## Round 0.3 · accepted · Accept

The Authors have addressed all the criticisms raised the reviewers and revised the manuscript accordingly. The paper is acceptable in the present form.

#